# Enhanced summertime ozone and SOA from biogenic volatile organic compound (BVOC) emissions due to vegetation biomass variability during 1981–2018 in China

**Jing Cao[1], Shuping Situ[2], Yufang Hao[3], Shaodong Xie[4], Lingyu Li[1]**

[1]College of Environmental Sciences and Engineering, Qingdao University, Qingdao 266071, China
[2]Foshan Ecological and Environmental Monitoring Station of Guangdong Province, Foshan 528000, China
[3]Laboratory of Atmospheric Chemistry, Energy and Environment Research Division, Paul Scherrer Institute/ETH, Villigen 5232, Switzerland
[4]State Key Joint Laboratory of Environment Simulation and Pollution Control, College of Environmental Sciences and Engineering, Peking University, Beijing 100871, China

Correspondence: Lingyu Li (lilingyu@qdu.edu.cn)

**Abstract.** Coordinated control of fine particulate matter ($PM_{2.5}$) and ozone ($O_3$) has become a new and urgent issue for China's air pollution control. Biogenic volatile organic compounds (BVOCs) are important precursors of $O_3$ and secondary organic aerosol (SOA) formation. China experienced a rapid increase in BVOC emissions as a result of increased vegetation biomass. We applied WRF-Chem3.8 coupling with MEGAN2.1 to conduct long-term simulations for impacts of BVOC emissions on $O_3$ and SOA during 1981–2018, using the emission factors extrapolated by localized emission rates and annual vegetation biomass. In summer of 2018, BVOC emissions are 9.91 Tg (in June), which lead to an average increase of 8.6 ppb (16.75% of the total) in daily maximum 8-h (MDA8) $O_3$ concentration and 0.84 μg m$^{-3}$ (73.15% of the total) in SOA over China. The highest contribution to $O_3$ is concentrated in the Great Khingan Mountains, Qinling Mountains, and most southern regions, while southern areas for SOA. Isoprene has the greatest contribution to $O_3$ while monoterpene has the largest SOA production. BVOC emissions have distinguished impacts in different regions. Chengdu-Chongqing (CC) region has the highest $O_3$ and SOA generated by BVOCs while Beijing-Tianjin-Hebei (BTH) region has the lowest. From 1981 to 2018, the interannual variation of BVOC emissions caused by increasing leaf biomass results in $O_3$ concentration increasing by 7.38% at an average rate of 0.11 ppb yr$^{-1}$, and SOA increasing by 39.30% at an average rate of 0.008 μg m$^{-3}$ yr$^{-1}$. Due to the different changing trends of leaf biomass by regions and vegetation types, $O_3$ and SOA show different interannual variations. Fenwei Plain (FWP), Yangtze River Delta (YRD), and Pearl River Delta (PRD) regions have the most rapid $O_3$ increment while the increasing rate of SOA in CC is the highest. BTH has the smallest enhancement in $O_3$ and SOA concentration. This study will help to recognize the impact of historical BVOC emissions on $O_3$ and SOA, and further provide the reliable scientific basis for the precise prevention and control of air pollution in China.

## 1 Introduction

In recent years, China suffers from more and more severe $O_3$ pollution with continuously increasing $O_3$ concentration in most regions (Liu et al., 2018a; Liu and Wang, 2020; Wang et al., 2017). On the other hand, although the ambient concentrations of $PM_{2.5}$ have decreased, its pollution is still severe (Fan et al., 2020; Li et al., 2019b; Silver et al., 2018; Zhai et al., 2019). $PM_{2.5}$ and $O_3$ are currently important atmospheric pollutants affecting air quality in urban and regional areas of China. Their coordinated prevention and control has become a new and urgent issue for China's air pollution control. Volatile organic compounds (VOCs) are precursors of $O_3$ and secondary organic aerosol (SOA) formation (Claeys et al., 2004; Hallquist et al. 2009; Kota et al. 2015), which can be emitted by biogenic and anthropogenic sources. On the global scale, up to 90% of total VOC emissions come from biogenic sources, including 99% from vegetation (Guenther et al., 1995). Therefore, the biogenic VOCs (BVOCs) in this study refer to the VOCs emitted by vegetation. BVOCs play a key role in the formation of secondary pollution due to their large emission and high activity, including isoprene, monoterpene, and sesquiterpene (Carslaw et al., 2010; Emanuelsson et al., 2013; Ng et al., 2008; Tasoglou and Pandis, 2015). Globally, BVOC emissions contribute about 20% and 76% to the $O_3$ and SOA formation, respectively (Hallquist et al., 2009; Wang et al., 2019). In China, BVOC emissions are estimated to be approximately 1.8 times that of anthropogenic VOCs (Li et al., 2016; Yang et al., 2021). The high BVOC emissions can lead to great increases by 29–49% in surface $O_3$ concentration in most southern urban areas (Li et al., 2018; Liu et al., 2018b; Situ et al., 2013; Wu et al., 2020) and contribute 70–75% to China's total SOA formation in summer (Hu et al., 2017; Wu et al., 2020). Therefore, it is essential to understand the emission characteristics of BVOCs and their impacts on the formation of $O_3$ and SOA for making effective policies of secondary air pollution control in China.

In the past four decades, China's BVOC emissions have increased by 55.38% due to biomass growth and climate change, of which biomass variability dominates the interannual variations in emission, which were reported in our previous studies (Li and Xie, 2014; Li et al., 2021). During 1981–2018, the leaf biomass of forests and crops increased by 118.63% and 316.83%, respectively, resulting in a continuous growth trend in BVOC emissions by 0.52 Tg $yr^{-1}$ averagely. The increasing BVOC emissions will affect their contributions to $O_3$ and SOA formation. Some studies investigated the impact of historical and future BVOC emissions on $O_3$ and SOA. Fu and Liao (2012) applied GEOS-Chem coupling with the Model of Emissions of Gases and Aerosols from Nature (MEGAN) using the default constant BVOC emission factors and annual leaf area index (LAI) data, and found that the interannual variations in BVOCs led to 2–5% differences in simulated $O_3$ and SOA in summer over years 2001–2006. Li et al. (2015) concluded that the BVOC emissions (including isoprene, α-pinene, and β-pinene) from the early 1990s to 2006 caused a 0.9–4.6 ppb increment in $O_3$ concentrations over the downwind areas in the PRD region, based on the annual land cover data. Sun et al. (2019) used MEGAN2.1 to calculate biogenic emissions which only include isoprene and monoterpenes and showed that their interannual changes caused by meteorology led to an increment of 0.3 ppb $O_3$ over central eastern China from 2003 to 2015. Liu et al. (2019) concluded that climate-driven BVOC emission changes would enhance $O_3$ and SOA concentrations by 0.90% and 7.33% in eastern China from 2015s to

2050s under future climate scenario RCP8.5, and reduce them by 0.80% and 6.50% respectively under scenario RCP4.5, simulated by the same emission factors and land cover data. However, the BVOC emission inventories applied in these studies may have high uncertainties because of the lack of localized emission factors and high-resolution land cover data, which is not conducive to the accurate evaluation for the contribution of BVOC emissions on $O_3$ and SOA formation. In addition, China experienced continuously increasing forest volume, while few studies focus on the impact of changes in BVOCs emissions caused by vegetation biomass variability over a longer time span. These make it difficult to accurately evaluate the impact of BVOCs on $O_3$ and SOA and guide the precise control of $O_3$ and $PM_{2.5}$. How to obtain the accurate historical contribution of BVOCs on $O_3$ and SOA based on localized accurate BVOC emissions is both an important issue and a great challenge.

In this study, firstly we updated the localized emission factor and detailed vegetation data to obtain more accurate BVOC emissions. The emission factors were extrapolated by localized emission rates and annual vegetation biomass at the provincial level. Then, based on the estimated historical emissions, their long-term impacts on $O_3$ and SOA during 1981–2018 were simulated through coupling MEGAN2.1 in WRF-Chem3.8. We aimed to investigate the impact of interannual BVOC emission variations on $O_3$ and SOA formation caused by vegetation biomass variability. The spatial variations in BVOC effects were discussed. Also, their contribution by different components was studied. It is expected that an accurate evaluation of BVOC contribution to $O_3$ and SOA can be obtained through improving the accuracy of emission inventory. Our study can help to better understand the role of vegetation biomass variability in the BVOC impact on $O_3$ and SOA in China in the past four decades.

## 2 Model and simulation

### 2.1 MEGAN

The MEGAN2.1 was utilized to estimate BVOC emissions in China at a spatial resolution of 36 km × 36 km (Guenther et al., 2012). It can calculate hourly emissions of 147 BVOC species, including isoprene, monoterpene, sesquiterpene, carbonyls, and other VOCs. The BVOC emissions simulated by MEGAN2.1 were the online inputs of WRF-Chem simulation.

Meteorological and vegetation data are inputs to drive MEGAN. MEGAN2.1 requires hourly weather variables to drive the calculation of hourly BVOC emissions. The hourly meteorological fields including temperature, downward shortwave radiation, wind speed, water vapor mixing ratio, pressure, and precipitation were simulated by the WRF model in this study. Temperature and radiation play key roles in BVOC emissions. We used the observed daily average temperature at 2411 sites in 2008 and 684 sites in 2018 in China to evaluate the reliability of the 2-m temperature (T2) simulated by WRF in this study. The observations were from the National Meteorological Data Center in China (http://data.cma.cn/). The simulated radiation was not evaluated because of a lack of available site observations. We conducted the statistical verification of meteorological variables, as shown in Table S1, including the average mean bias (MB), mean absolute error (MAE), and root-mean-square

error (RMSE). The results show that the WRF simulation is considered reasonable for driving MEGAN. The vegetation data includes gridded fraction of plant functional types (PFTs), leaf area index (LAI), and PFT-specific emission factors. The new detailed PFT classification and distribution were developed from the Vegetation Atlas of China (1:1,000,000) at a high resolution of ~ 250 m, including 82 PFTs of forests, crops, grasses, and shrubs. The atlas was compiled based on multi-year vegetation survey data and related researches over 30–40 years, which provides the more detailed distribution of vegetation

in China. Previous studies typically included a coarse vegetation classification that is based on a less-detailed vegetation distribution (Gao et al., 2019; Klinger et al., 2002; Wang et al., 2007). And the MEGAN2.1 defined 15 vegetation types by default (Guenther et al., 2012). For LAI, the MODIS LAI data was used.

Emission factors were extrapolated by the leaf-level emission rates and leaf biomass using the canopy environment model described in MGEAN2.1. In our previous studies on BVOC emission inventories, a large number of observations

from China and other countries were summarized to obtain more accurate basal emission rates using a theoretically effective statistical approach (Li et al., 2020). The dataset contained isoprene and monoterpene emission rates of 192 plant species/genera including the dominant forest tree and crop species, and grass and shrub genera, which were expected to be more accurate and localized. In previous studies, traditional emission categories were used to determine emission rates (Guenther et al., 1994; Klinger et al., 2002; Simpson et al., 1999; Wang et al., 2007), which usually utilized coarse categories

and resulted in high uncertainty. The vegetation species/type specific leaf biomass at the province level originated from statistics of vegetation volume and production using biomass apportion models, which revealed differences among plant species and regions comparing with previous studies (Li et al., 2013, 2020; Li and Xie, 2014). Previous studies usually applied an average value for each vegetation class, such as broadleaf trees, needleleaf trees, crops, and grasses, without revealing their differences among regions and plant species (Klinger et al., 2002; Wang et al., 2007). The statistics on forest

volumes were from the National Forest Inventory of China. Eight forest inventories for the periods of 1977–1981, 1984–1988, 1989–1993, 1994–1998, 1999–2003, 2004–2008, 2009–2013, and 2014–2018 were used. The last year of each period were selected to simulate the historical emissions and concentration of $O_3$ and SOA during 1981–2018. Crop productions were from China Statistical Yearbook for the year 1981, 1988, 1993, 1998, 2003, 2008, 2013, 2018. Grass productions were from the Grassland Resource Data of China. Consequently, the extrapolated emission factors vary over years. After emission

factor calculation of total monoterpene, the emission factors for each monoterpene and sesquiterpene species were allocated from total monoterpene based on the global average emission factor proportions described in MEGAN2.1. For the emission factors of other VOCs, the default MEGAN emission factors were used.

**2.2 WRF-Chem**

The chemistry version of the WRF model (WRF-Chem3.8) was used in this study to simulate the concentration of

125 surface $O_3$ and SOA (Grell et al., 2005). WRF-Chem realizes the online coupling of meteorological models and chemical models, considering the emission, transport, diffusion, dry and wet deposition, photolysis, meteorological chemistry, and aerosol chemical processes of pollutants, which have been widely used to make the on-line calculations of meteorology and

chemistry. A large number of global and regional air pollution studies widely apply it to simulate secondary pollutants, and the verification results show that it can well reproduce the observed pollutant concentrations (Gupta and Mohan, 2015;
Hoshyaripour et al., 2016; Li et al., 2018; Situ et al., 2013; Wu et al., 2018).

One nested domain centered at 34.53°N and 108.92°E was adopted, with a 36-km horizontal resolution covering the whole area of the China region. Main parameterization options for physical and chemical schemes of the model setup are listed in Table S2. The United States National Centers for Environmental Prediction (NCEP)/Department of Energy (DOE) Reanalysis II data at $1° \times 1°$ were used as the initialization field and boundary conditions for WRF, which are updated every
6 h (00, 06, 12, 18 UTC). The NOAA/ESRL RACM (Stockwell et al., 1997) gas phase chemistry scheme and the volatility basis set (VBS) (Donahue et al., 2006) aerosol chemistry module were selected in this study. The RACM consists of 77 chemical species and considers 237 reactions, including relatively detailed organic chemistry which considers the oxidation mechanism for BVOCs (e.g. isoprene, α-pinene, β-limonene, etc.). In VBS, a unified set of saturated vapor pressure is used, and the coupling matrix of gas phase and condensed phase is established to describe the photochemical multi-generation
oxidation and gas-particle partition process. This approach can better represent multi-generation oxidation of BVOCs in the gas phase and their aging processes in the aerosol phase (Hu et al., 2017). The photolysis scheme of Fast-J (Wild et al., 2000) was selected because it can better compute photolysis rates from the predicted $O_3$, aerosol, and cloud profiles.

The observed daily maximum 8-h (MDA8) $O_3$ and daily average $PM_{2.5}$ concentrations at 1588 sites in June 2018 in China were applied to evaluate the WRF-Chem simulations in the control run (as listed in Table 1) in this study. The
observations were from the daily updated national air quality released by the China National Environmental Monitoring Centre (http://www.cnemc.cn/). The verification statistics are shown in Table S1. Notably, $PM_{2.5}$ had no systematic bias between the observation and simulation, while the model predicted $O_3$ concentrations were lower than measurements. The errors could be mainly attributed to the anthropogenic emissions data used in this study as described in Section 2.3.

**2.3 Simulation setups**

In this study, June was selected as the simulated period to investigate the effects of summertime BVOC emissions on $O_3$ and SOA generation. Seven simulation scenarios were set up to investigate the effects of BVOC emissions by each compound category and historical emission variation. The details of simulation setup are listed in Table 1. We focused on the contribution of historical BVOC emissions caused by biomass variability, so the anthropogenic emissions in all scenarios were fixed using the MIX Asian anthropogenic source emission inventory (Li et al., 2017).

In the BASE scenario, one-month long simulation using anthropogenic emissions and total BVOC emissions in June of 2018 was conducted, as the control run for this study. Excluding all the BVOC emissions, the BIO scenario was set to simulate the $O_3$ and SOA concentration to quantify the impacts of total BVOC emissions compared with the results of BASE scenario (BASE-BIO). Different compound categories have different contributions to the formation of secondary pollutants. Thus, scenarios "ISOP", "MTP", "SQT", and "ISOPRENOID" were designed to simulate the impacts of isoprene,
monoterpene, sesquiterpene, and isoprenoid (total of the above three categories) on $O_3$ and SOA, by excluding their

emissions, respectively. For these simulations, the meteorology of 2018 was used to drive MEGAN to estimate biogenic emissions in June 2018. Scenario HISTORY was an interannual comparison simulation to estimate the impacts of historical BVOC emissions caused by the change of vegetation leaf biomass. It was run by using annual emission factors extrapolated from emission rates and annual leaf biomass during 1981–2018. For the meteorology, the fixing set of a mid-year 2008 over 1981–2018 were used for all the HISTORY simulations. To explore the impacts of interannual BVOC emission variations caused by vegetation biomass variability, influences of annual meteorology on BVOC emissions and formation of secondary air pollutants were not considered.

**Table 1.** Description of different model simulations in this study.

| Simulation | Anthropogenic emissions | BVOC emissions | BVOC Emission factor | Meteorology |
|---|---|---|---|---|
| BASE | All | All BVOCs | Year 2018 | Year 2018 |
| BIO | All | No BVOCs | - | Year 2018 |
| ISOP | All | No isoprene | Year 2018 | Year 2018 |
| MTP | All | No monoterpene | Year 2018 | Year 2018 |
| SQT | All | No sesquiterpene | Year 2018 | Year 2018 |
| ISOPRENOID | All | No isoprenoid | Year 2018 | Year 2018 |
| HISTORY | All | All BVOCs estimated with annual emission factors | Years 1981–2018 | Year 2008 |

# 3 Results and discussion

## 3.1 BVOC emission

### 3.1.1 Spatial distributions of BVOC emissions

The total BVOC emissions in China estimated by MEGAN2.1 are 9.91 Tg in June 2018, of which isoprene, monoterpene, sesquiterpene, and other BVOCs account for 64.21%, 10.58%, 2.12%, and 23.09%, respectively. As shown in Fig. 1, BVOC emissions show significant spatial variations with the highest emissions in the Changbai Mountains, Greater Khingan Mountains, Qinling Mountains, the southeast and southwest China forest regions, and Hainan and Taiwan provinces, and the lowest in the Qinghai-Tibet Plateau and southern Xinjiang province.

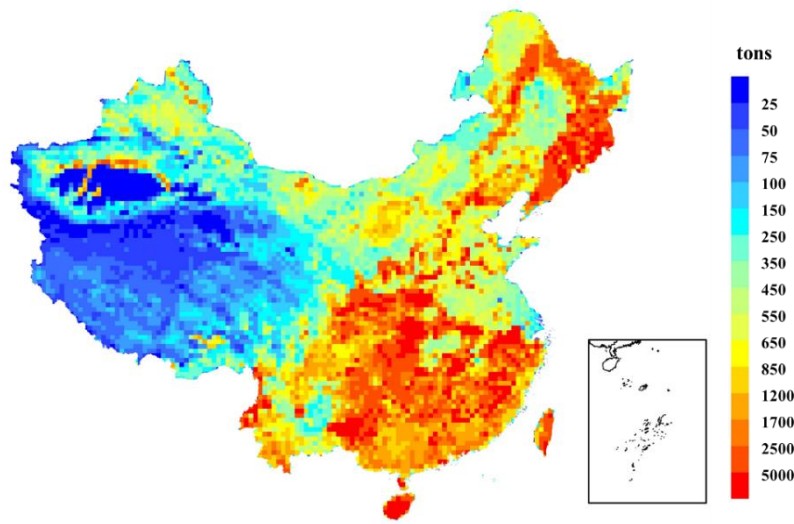

**Fig. 1.** Spatial variations in BVOC emissions in June 2018.

The emission simulations were validated by using the formaldehyde (HCHO) concentration and measurements of BVOC emission flux. Isoprene is the main compound in BVOC species, accounting for nearly half of total BVOC emissions in China. It undergoes chemical and photochemical reactions in the atmosphere, and the oxidation product is mainly HCHO (Bai and Hao, 2018; Orlando et al., 2000). In forest areas and in summer, biogenic isoprene is the dominant source of HCHO,

so satellite HCHO column concentration is widely used to constrain isoprene emissions (Opacka et al., 2021; Palmer et al., 2003; Stavrakou et al., 2018; Wang et al., 2021; Zhang et al., 2021). In this study, we used the HCHO vertical column detected by Ozone Monitoring Instrument (OMI) to validate the spatial variability of isoprene estimates. The monthly OMI HCHO data from the EU FP7 project QA4ECV product (Quality Assurance for Essential Climate Variables; http://www.qa4ecv.eu) was used in this study. The result of the statistical analysis with a confidence interval of 99%

indicates that the monthly averaged OMI HCHO vertical column in June 2018 is significantly correlated to the model-estimated isoprene emissions. The flux measurements of BVOCs conducted in China were collected (Bai et al., 2015, 2016, 2017). The details of these measurements are provided in Table S3, including the location of sampling sites and observed BVOC emission fluxes. The gridded BVOC emission estimated by MEGAN were extracted where the flux measurement sites were located to do the comparison (Fig. S1). The modeled fluxes of BVOCs in this study capture the

spatial variability of observations better with a correlation coefficient of 0.84. But the estimation is higher than measurement with an average mean bias of 1.11 mg m$^{-2}$ h$^{-1}$ in these sampling sites, mainly because of the differences in time between them.

### 3.1.2 Influence of leaf biomass variability

The leaf biomasses increased from $378.35 \times 10^{12}$ g in 1981 to $1107.16 \times 10^{12}$ g in 2018 at an average rate of $17.97 \times 10^{12}$ g yr$^{-1}$. Among them, the forest and crop leaf biomass increased from $237.10 \times 10^{12}$ to $518.38 \times 10^{12}$ g and from $141.25 \times 10^{12}$ to $588.79 \times 10^{12}$ g, respectively, totally increasing by 192.63%. The spatial distribution of interannual variations in leaf biomass is presented in Fig. S2. The increase of leaf biomass is most significant in Great Khingan, Changbai Mountains, North China Plain, south and southwest China. This is mainly due to the increased stock of broadleaf and coniferous forests as a result of afforestation. Northern Qinghai-Tibet area and Northwest China have a relatively high grass cover rate but insignificant increase in leaf biomass of vegetation. It is because that the grass biomasses were the same over the historical simulations due to lacking of data.

Due to the increased volume and production of vegetation, the total BVOC emissions increased by 58.66% at average rates of 96.64 Gg yr$^{-1}$, of which isoprene, monoterpene, sesquiterpene increased by 108.57%, 38.17%, and 33.35% at average rates of 11.10, 0.99, and 0.17 Gg yr$^{-1}$, respectively. Isoprene emissions increased more rapidly over the past 40 years, which is primarily due to the greater increase in the biomass of broadleaf trees, which have the highest isoprene emission rates. Monoterpene and sesquiterpene increased at a lower rate because the increase of leaf biomass of conifers is relatively small. Fig. 2 shows the spatial distribution of interannual variations in BVOC emissions caused by the changing leaf biomass. Since the needleleaf and broadleaf trees tend to have a higher emission potential than grass or crop (Guenther et al., 2012), their wide distribution and the substantial increase in biomass result in the largest interannual variability of BVOC emissions in the Great Khingan, Changbai Mountains, North China Plain, Central and Southern China, and Hainan Province. However, the emission of BVOCs in the northwest and southern coastal areas has decreased.

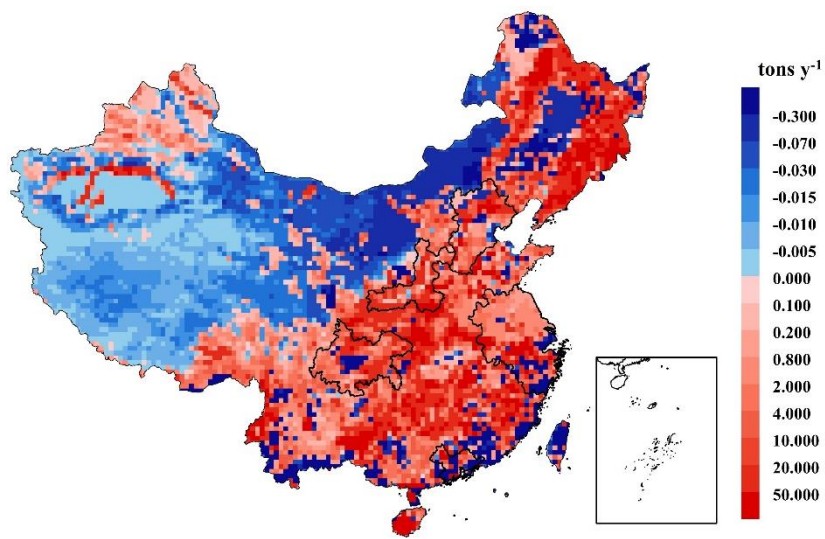

**Fig. 2.** Spatial distribution of interannual variations in BVOC emissions caused by leaf biomass changes.

 **3.2 Impacts of BVOC emissions on O$_3$ and SOA**

**3.2.1 O$_3$**

BVOC emissions generate additional O$_3$ and contribute 8.61 ppb MDA8 O$_3$ over China on average in the 2018 summer. Their nationally averaged contribution is 16.75%. There are distinct differences spatially in the impact of BVOC emissions. The spatial distribution of the difference in MDA8 O$_3$ concentrations between simulations with and without BVOC emissions in summer 2018 is shown in Fig. 3. BVOC emissions lead to the increase of O$_3$ concentration in most regions with the highest contribution of > 30 ppb. The high contribution is concentrated in southern China, which is mainly caused by their higher BVOC emissions due to higher temperature and vegetation biomass (Li et al., 2020). Their impact ranges from 10.50 to 77.17 ppb, contributing 36.20–70.83% to the total O$_3$ formation, respectively. Because most land cover in western China, including Qinghai-Tibet Plateau and Xinjiang province, are sparse or herbaceous vegetation with low emission potential, O$_3$ is less affected by BVOC emission. In addition, it is interesting to note that the extremely low contribution in some areas of the Bohai Rim in eastern region is inconsistent with the high BVOC emissions. This is likely because O$_3$ is more sensitive to anthropogenic emissions.

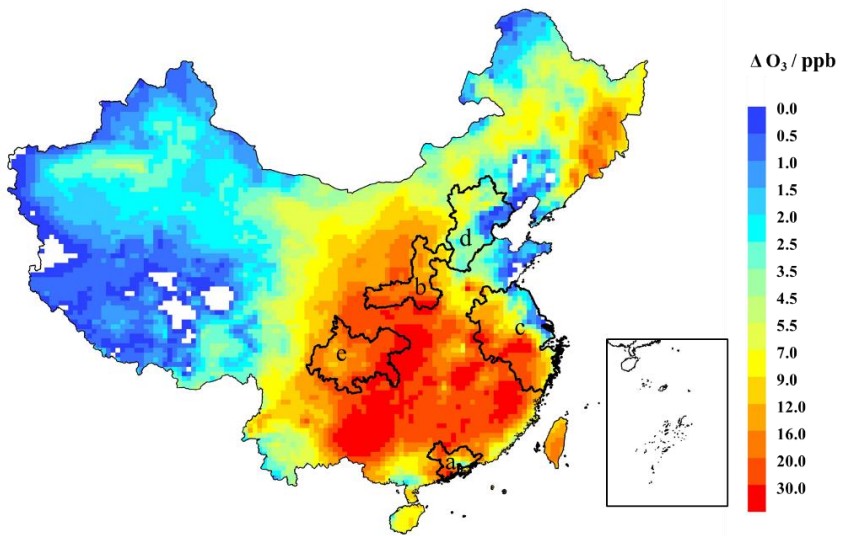

**Fig. 3.** Spatial variations in impact of BVOC emission on MDA8 O$_3$ concentration. The key regions include (a) Pearl River Delta (PRD), (b) Fenwei Plain (FWP), (c) Yangtze River Delta (YRD), (d) Beijing-Tianjin-Hebei (BTH), and (e) Chengdu-Chongqing (CC).

The spatial pattern of estimated MDA8 O$_3$ impacted by BVOC emissions differs from the spatial distribution of BVOC emissions mainly because of the variability of the nonlinear response relationship between O$_3$ formation and precursors. As the important precursors, VOCs and NO$_x$ react in the presence of hydroxyl (OH) and hydroperoxyl (HO$_2$) radicals to create

$O_3$. The $O_3$ formation is expected to be affected by the different levels of $O_3$ precursors in different land use functional areas. According to different VOCs/$NO_x$ ratio, $O_3$ formation regimes can be classified into VOC-limited (VOC-sensitive), transition, and $NO_x$-limited ($NO_x$-sensitive) regimes (Lu et al., 2019; Wang et al., 2008). From the spatial distribution of the BVOC effect (Fig. 3), the surface $O_3$ is sensitive to BVOC emissions in most regions in China which can furtherly indicate

they are usually VOCs-limited. It confirms the conclusion made by Lu et al. (2019), Lyu et al. (2016), and Tan et al. (2018) that the VOCs-limited regime is dominant in southern China. Comparing with the spatial distribution of BVOC emissions (Fig. 1), the areas with high BVOC emissions usually have a higher contribution to $O_3$. Because the dense population leads to a large number of $NO_x$ emitted by human activities, $NO_x$ is saturated with the formation of $O_3$ which is more sensitive to VOC emissions. Therefore, the higher BVOC emissions usually cause greater contribution to $O_3$ in these areas. In the

VOC-limited regime, the reduction of VOC emissions reduces the chemical production of organic radicals ($RO_2$), which in turn lead to decreased cycling with $NO_x$ and consequently lower concentration of $O_3$ (Jin and Holloway, 2015; Milford et al., 1989). To decrease BVOC emissions by planting plants with low emission potential may contribute to $O_3$ pollution control. Notably, both northeastern and southern regions have the highest BVOC emissions, but their contributed $O_3$ differ much, which indicates that $O_3$ formation in the south is more sensitive to VOCs than in the northeast. Hainan province also has

higher BVOC emissions but relatively lower contribution to $O_3$.

Five key regions with stronger economy, larger population, and higher $O_3$ pollution are selected to further investigate the regional variability in the impact of BVOC emissions on $O_3$, which are BTH, FWP, PRD, YRD, and CC regions (Table 2). Their BVOC effects were more significant than the national average except for the BTH region. These four key areas are populous, economically developed, and power grids are dense. The high concentration of $NO_x$ emissions may lead to

260 saturation of $NO_x$ and excessive consumption of OH, so ozone generation is more sensitive to VOC emissions (Jin and Holloway, 2015). The contribution is the largest in CC region with 23.29 ppb $O_3$ increment, which accounts for 44.98% of the total $O_3$ concentration, although its BVOC emissions are relatively lower among the five regions. It is mainly due to its low concentration of other precursors including $NO_2$ and CO in summer and low topography that is favorable for the accumulation of $O_3$ (Cao et al., 2018). The difference in MDA8 $O_3$ with and without BVOC emissions shows a small gap

between FWP and PRD regions, both of which are ~ 19 ppb, but the contribution to total $O_3$ in PRD (41.83%) is 1.17 times those in FWP (35.81%), which can indicate higher contribution by the anthropogenic VOC emissions in FWP. In addition, FWP has higher $O_3$ concentrations in summer than PRD (Li et al., 2019a; Wang et al., 2019; Xie et al., 2021). The BVOC emissions in BTH are the lowest among the five regions, so the impact on MDA8 $O_3$ is the least significant which is 4.10 ppb and 15.84% of the total $O_3$. This can be explained by the greater contribution of anthropogenic emissions to surface $O_3$

caused by manufacturing, electricity production, and transportation.

### 3.2.2 SOA

BVOCs undergo a series of atmospheric degradation processes to produce oxidation products, which may contribute to the formation and growth of SOA. This process is affected by concentration of precursors and meteorological conditions. In

summer, high biogenic emissions combined with enhanced photochemical levels caused by high temperature and strong solar radiation lead to the peak of SOA production in this season (Kelly et al., 2018). From our simulations, SOA produced by BVOCs (BSOA) plays an important role in SOA production, accounting for 73.15% of the total concentration, which is in good agreement with previous studies that considered BSOA as a major contributor (Hu et al., 2017; Wu et al., 2020). The BVOC emissions generate additional ~ 0.8 μg m$^{-3}$ SOA on average with a gridded maximum of 4.12 μg m$^{-3}$. Figure 4 shows the spatial distribution of BSOA contribution in June 2018. All the country has increases in SOA concentration after considering BVOC emissions comparing with simulation without them. Generally, spatial variability in BVOC contribution to SOA is corresponding with that in BVOC emissions. Southern China has a higher contribution than northeastern areas although they both have the highest BVOC emissions in summer. It is mainly because the southern area has higher emissions of monoterpenes that contribute most to SOA generation. Notably, the hotspots of BVOC contribution to SOA with > 2 μg m$^{-3}$ (accounting for > 70% of the total SOA) are mainly distributed in Sichuan Basin, Northeast Yunnan, and Northwest Guizhou, which may be because the high level of BVOCs in southern China can be transported to these regions due to the influence of the southeast monsoon in the summertime and accumulate under the barrier of the western plateau (Wang et al., 2018). The sensitivity of SOA to BVOC emission is the lowest in the western Qinghai-Tibet Plateau and southwestern Xinjiang, where the contribution of BVOCs is only about 0.15 μg m$^{-3}$, lower than 60% of the total produced SOA. This is mainly because of their lower biogenic emissions contribute.

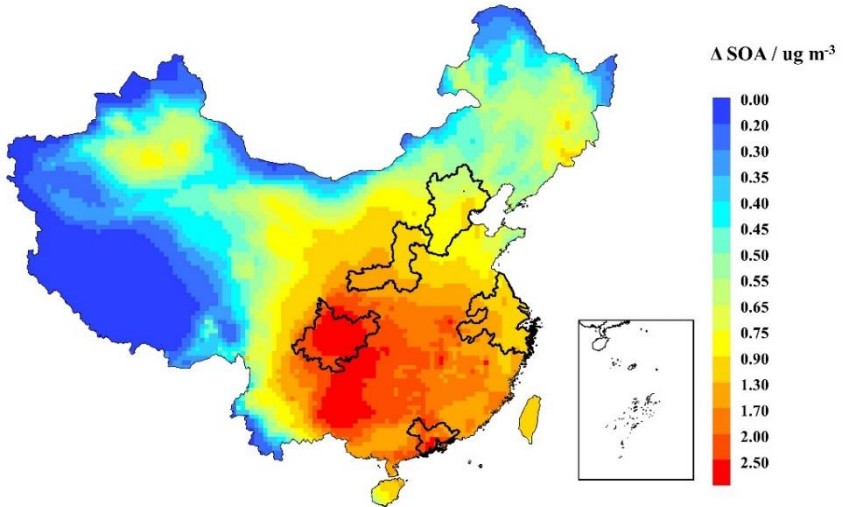

**Fig. 4.** Spatial variations in impact of BVOC emission on SOA concentration.

The impacts of BVOC emissions on SOA differ much among the five key regions (Table 2). The results indicate that SOA production is most sensitive to BVOC emissions in CC, followed by PRD, FWP, and YRD, and the least in BTH. Among the five regions, CC has the lower BVOC emissions but the strongest impacts on SOA, with an additional 2.51 μg m$^{-3}$ SOA and contributing 84.27% to the total, which characteristic coincide with impacts on O$_3$. This can be explained by

the widely distributed forest in the southwest forest region and the topography that is beneficial to the accumulation of BVOCs and SOA. The BVOC emissions of 462.92 Gg in PRD generate 1.96 µg m$^{-3}$ SOA, accounting for 74.63% of the total SOA. YRD and FWP regions have similar emission budgets and additional SOA generation by BVOCs but different proportions of BSOA in the total. The contribution of BSOA in FWP (77.75%) is greater than that in YRD (69.12%), which indicates higher contribution of anthropogenic VOCs in YRD. BTH has the lowest production of BSOA with 0.74 µg m$^{-3}$, which is lower than the national average, mainly due to its lowest biogenic emissions among the five regions.

### 3.2.3 BVOC component contribution

Table 2 shows the contributions of different BVOC categories to MDA8 O$_3$ and SOA. Isoprene, monoterpene, and sesquiterpene contribute 7.01, 1.17, and 0.16 ppb O$_3$ and 0.25, 0.52, and 0.22 µg m$^{-3}$ SOA over China, respectively. Among the emission dominated categories, including isoprene, monoterpene, and sesquiterpene, isoprene has the largest contribution to O$_3$ production. Owing to its highest emissions and reactivity with various oxidants, isoprene plays an important role in the formation of O$_3$ (Wennberg et al., 2018). Sesquiterpene emission is the lowest, resulting in the smallest contribution to O$_3$. As key precursors of SOA formation, monoterpene emissions are only 1/6 of isoprene, but their contribution to SOA is the highest, about two times contribution by isoprene. The reaction of monoterpenes with atmospheric oxidants such as O$_3$, OH, and NO$_3$ is considered to be an important reason for the SOA formation from them (Hoffmann et al., 1997; Mutzel et al., 2016; Watne et al., 2017). The contribution of isoprene to SOA is relatively lower because isoprene mainly reacts with OH radicals to form SOA while the reactions with other oxidants in the atmosphere (including O$_3$ and NO$_3$) contribute little to SOA production (Claeys et al., 2004; Henze and Seinfeld, 2006; Ruppert and Becker, 2000). Sesquiterpene has much lower emissions than isoprene, but their generated SOA are equivalent, indicating the higher SOA yield of sesquiterpene (Hoffmann et al., 1997). From the simulation of scenario "ISOPRENOID", which excluding the emissions of isoprene, monoterpene, and sesquiterpene together, the additional generation of O$_3$ and SOA by isoprenoid are not a linear combination of those by each category. The contributions by isoprenoid are usually larger than the sum of those by each category. It can be concluded that there is a complex nonlinear relationship between BVOCs and the formation of O$_3$ and SOA.

**Table 2.** Emissions of each BVOC category and their corresponding contribution to MDA8 O$_3$ and SOA concentration in the five key regions of China in June 2018.

|  | BVOC category | China | BTH | FWP | PRD | YRD | CC |
|---|---|---|---|---|---|---|---|
| Emission | Isoprene | 636.28 | 15.66 | 30.29 | 34.74 | 53.43 | 24.50 |
| (10$^4$ tons) | Monoterpene | 104.81 | 2.77 | 2.16 | 5.09 | 4.38 | 11.59 |

|  |  |  |  |  |  |  |
|---|---|---|---|---|---|---|
|  | Sesquiterpene | 20.98 | 0.64 | 0.32 | 1.27 | 1.42 | 1.65 |
|  | Total BVOCs | 990.91 | 29.23 | 40.67 | 46.29 | 74.18 | 51.32 |
| Contribution to MDA8 $O_3$ (ppb) | Isoprene | 7.01 | 3.42 | 18.01 | 16.81 | 12.55 | 20.49 |
|  | Monoterpene | 1.17 | 1.74 | 0.32 | 4.07 | 2.92 | 6.89 |
|  | Sesquiterpene | 0.16 | 0.93 | -1.36 | 1.66 | 1.06 | 2.88 |
|  | Isoprenoid | 7.77 | 2.94 | 17.43 | 16.34 | 14.23 | 24.55 |
|  | Total BVOCs | 8.61 | 4.10 | 18.94 | 18.74 | 13.40 | 23.29 |
| Contribution to SOA ($\mu g\ m^{-3}$) | Isoprene | 0.25 | 0.20 | 0.53 | 0.95 | 0.63 | 0.91 |
|  | Monoterpene | 0.52 | 0.45 | 0.72 | 1.21 | 0.62 | 1.59 |
|  | Sesquiterpene | 0.22 | 0.21 | 0.26 | 0.49 | 0.31 | 0.75 |
|  | Isoprenoid | 0.84 | 0.78 | 1.30 | 1.96 | 1.29 | 2.52 |
|  | Total BVOCs | 0.84 | 0.74 | 1.29 | 1.96 | 1.27 | 2.51 |

For all the five regions, isoprene has both the largest emissions and the highest contribution of $O_3$ generation, followed by monoterpene and sesquiterpene. $O_3$ in CC, FWP, and PRD are greatly impacted by isoprene, where the impacts are on average 20.49, 18.01, and 16.81 ppb, respectively. Among them, CC has the lowest isoprene emissions but the concentration of $O_3$ generated by isoprene is the highest which mainly because of the low concentration of other precursors. Isoprene emissions are the greatest in PRD. For the impact of monoterpene on $O_3$, CC contributes the most because it has higher

monoterpene emissions. Monoterpenes are the largest contributor to SOA production among the key regions except YRD. In YRD, the generated SOA by isoprene and monoterpene are equivalent although their emissions distinguish. These also indicate the different SOA yields of isoprene and monoterpene. SOA contributed by isoprene is also higher in CC and PRD regions due to their absolute advantage in topography and emission, respectively. BTH and FWP have similar monoterpene emissions, but their generated SOA differ much due to the different ratios of the different BVOC components emissions.

Comparing with isoprene and monoterpenes, the contribution of sesquiterpene to SOA is small.

### 3.3 Interannual variability

### 3.3.1 $O_3$

     The concentration of $O_3$ enhances from 48.82 ppb to 52.43 ppb at an average growth rate of 0.11 ppb $yr^{-1}$ during 1981–2018. At the 95% confidence level, the increasing trend for $O_3$ is significant. But the interannual variability in $O_3$

distinguishes regionally. Figure 5 shows the spatial distribution for changes of MDA8 $O_3$ concentration due to the changed BVOC emissions caused by vegetation leaf biomass variability in the past 40 years. The strongest enhancement of $O_3$ from interannual variation of BVOC emissions are mainly distributed in most of the southern areas, where have an average annual rate of > 0.2 ppb and some > 0.4 ppb. In these areas, the increase in biomass of broadleaf trees and needleleaf trees with high emission potential have contributed to the increased BVOC emissions and $O_3$ concentration as a result. The enhancement of $O_3$ in the area of the Changbai Mountains is also large mainly due to the increased broadleaf tree biomass. The weakest increments are found mainly in Xinjiang, southwestern Yunnan Province, and Taiwan. In Xinjiang Province, there is a greater distribution of vegetation with lower emission potentials, including crops and shrubs, which has a small change in leaf biomass. In the past four decades, the leaf biomass in Xinjiang increased by $17.55 \times 10^{12}$ g, accounting for only 2.41% of the total growth in China. The tropical forest in the southwest of Yunnan province has declined volume because of the commercial deforestation and expanded plantations of economic trees.

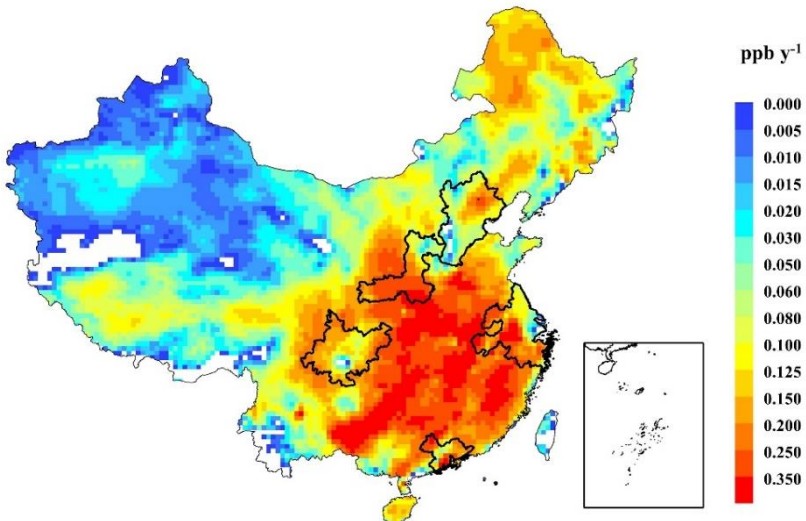

**Fig. 5.** Spatial distribution of interannual variations in $O_3$ simulated using annual BVOC emission factors.

Although the overall positive influence of historical BVOC emissions on MDA8 $O_3$, different trends are shown during different periods within 1981–2018. In the 1980s, due to the demand for social development, urban land continued to expand, resulting in massive deforestation. The decrease of forest biomass led to a decrease in BVOC emissions, resulting in little changes in $O_3$ concentration (only 0.03 ppb) from 1981 to 1988. In order to offset the deterioration of the ecological environment caused by a large number of deforestation, large-scale afforestation activities have been carried out all over the country, and forest biomass has been increasing. Most of the planted species selected for this ecological project were the broadleaf trees with high emission potential (Klinger et al., 2002). During 1988 and 2018, the forest volume and crop production increased by 6.60 billion $m^2$ and 1487.25 megatons, respectively. As a result, the vegetation leaf biomass

continued to grow by 95.89% that led to 3.58 ppb O₃ enhancement by 6.88%. Among the annual growth rates, the growth was the biggest in 2003 with an increase of 1.99% comparing with 1998.

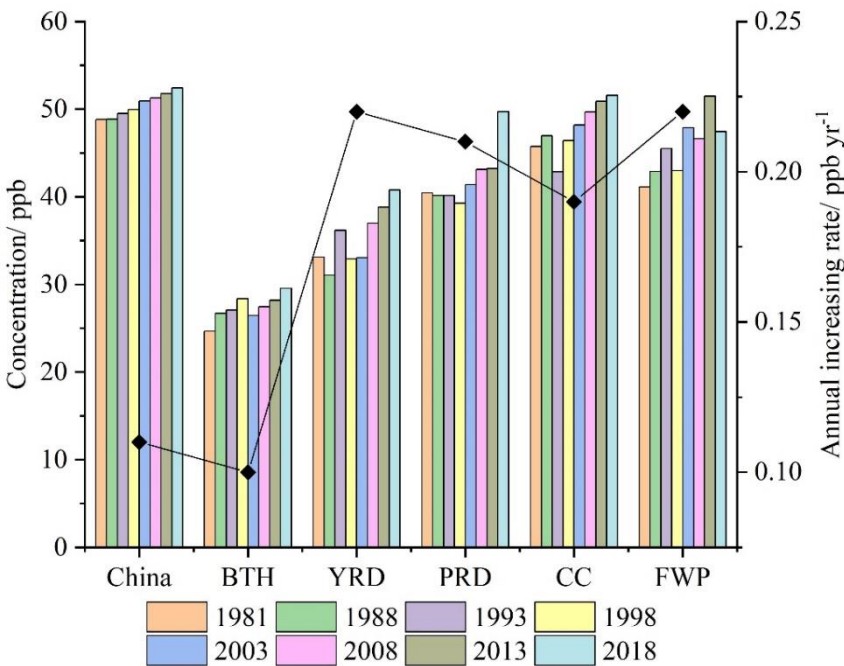

**Fig. 6.** The interannual changes of O₃ in China and the key regions.

The interannual variations in O₃ have significant differences among the five key regions (Fig. 6). But the overall growth trend can be found. From 1981 to 2018, MDA8 O₃ concentration in BTH, YRD, PRD, CC, and FWP increased by 19.87%, 23.24%, 22.91%, 12.71%, and 15.44% at average rates of 0.10, 0.22, 0.21, 0.19, and 0.22 ppb yr$^{-1}$, respectively. They have
O₃ increments of 4.90, 7.70, 9.26, 5.81, and 6.35 ppb, respectively. As the surface O₃ in urban areas is more sensitive to VOCs, the increment of O₃ concentration caused by BVOCs emissions in all the regions are greater than the national average. FWP, YRD, and PRD regions have the highest increasing rates annually which may be because of the rapid increase in leaf biomass of broadleaf trees with high isoprene emission potential. At the end of the last century, the implementation of reform and opening up led to the continuous acceleration of the level of urbanization and the reduction of forest and arable land
biomass and eventually resulting in the fluctuating decline of MDA8 O₃ in YRD and PRD during 1981–2003. The rapid growth of MDA8 O₃ in PRD led to a greater average annual growth rate of MDA8 O₃ in PRD greater than that in YRD from 1981 to 2018. The lowest growth rate occurs in BTH. Due to the Three North Shelterbelt System Project, the area coverage of natural forest in the north of BTH has more than tripled than that in 2003, resulting in an increase in leaf biomass and BVOC emissions (Ma et al, 2019). However, because of the rapid urbanization in Beijing, a large amount of forest has been
converted to urban construction land, and the reduction of BVOC emission related to losses of trees may offset part of their increase associated with rising coverage and volume in the surrounding areas.

### 3.3.2 SOA

The historically varied BVOC emissions caused by changes in leaf biomass had a significant influence on SOA formation in China from 1981–2018 according to our simulation. The national SOA enhanced at an annual rate of 0.01 μg m$^{-3}$, showing a significantly increasing trend ($p < 0.05$). As shown in Fig. 7 that depicts the spatial distributions of interannual changes in SOA, most of southern China shows a significantly increasing trend with an average growth rate higher than 0.02 μg m$^{-3}$, even > 0.03 μg m$^{-3}$ in some areas. It is mainly because of the continuous expansion of vegetation coverage and increase of biomass which results in enhanced BVOC emissions. Monoterpene is the biggest contributor to BSOA, as described in section 3.2.2. The spatial distribution of monoterpene emissions is consistent with that of needleleaf trees. The leaf biomass of needleleaf trees increased from 118.68 × 10$^{12}$ g to 212.04 × 10$^{12}$ g during 1981–2018. Needleleaf trees are densely distributed in southern China, including Masson pine, spruce, and hemlock with high monoterpene emission rates (Li et al., 2020). The highest average annual growth rates of SOA occur in the intersecting area of southwest and southeast China,and then the growth rate gradually declines to the surrounding areas. This can be partly attributed to the spatial distribution of needleleaf trees and the increasing volume of needleleaf trees in the southwest forest area. The lowest growth is found in the northwestern areas, southwest Yunnan, and Hainan provinces. Some of these areas show negative growth. In northwestern areas, BVOCs especially monoterpene emissions experienced the weakest growth and some even decreases. Although the ecological shelterbelt projects were conducted in northwest China, broadleaf tree species were mostly planted with lower monoterpene emissions. Because of the tropical climate in southwestern Yunnan and Hainan, needleleaf trees with greater monoterpene emission potential are less distributed and the change of leaf biomass is small, which led to small or even negative changes in the growth of SOA.

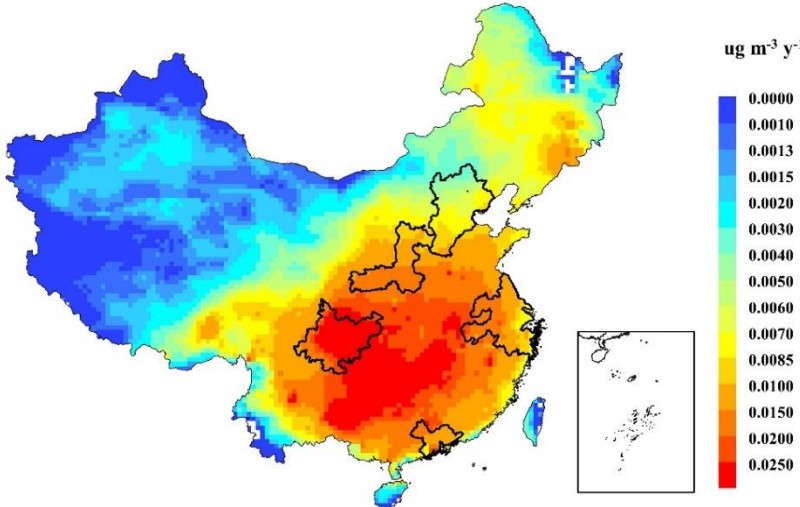

**Fig. 7.** Spatial distribution of interannual variations in SOA simulated using annual BVOC emission factors.

The national average SOA in 2018 is 1.06 μg m$^{-3}$, 39.30% higher than that in 1981. In the past 40 years, China has an overall growth for the SOA from BVOCs, and more rapid growth in 2003 and 2018. The historical SOA concentrations and their changes in BTH, YRD, PRD, CC, and FWP regions are presented in Fig. 8. They underwent the SOA increments by 31.08%, 53.21%, 37.97%, 54.61%, and 54.57% during 1981–2018, at average rates of 0.007, 0.014, 0.013, 0.027, and 0.014 μg m$^{-3}$ yr$^{-1}$, respectively. CC is the region with the largest annual increasing rate since it has the more significantly enhanced leaf biomass of needleleaf trees and crops which contribute much to monoterpene emissions. In BTH, SOA annual growth rate is the lowest and lower than the national average rate. It can be attributed to the obvious decrease of leaf biomass with 17.57 × 10$^{12}$ g from 1998 to 2008. With a similar overall growth rate, however, YRD and PRD have different interannual variability. YRD experienced two stages of increasing before and after 2003. In PRD, SOA showed a striking growth during the last five years owing to the increase of leaf biomass by up to 4.21 × 10$^{12}$ g. To sum up, there are significant increases in SOA in the five key regions, but they have marked differences in the regularity of growth due to the different changes of leaf biomass over time.

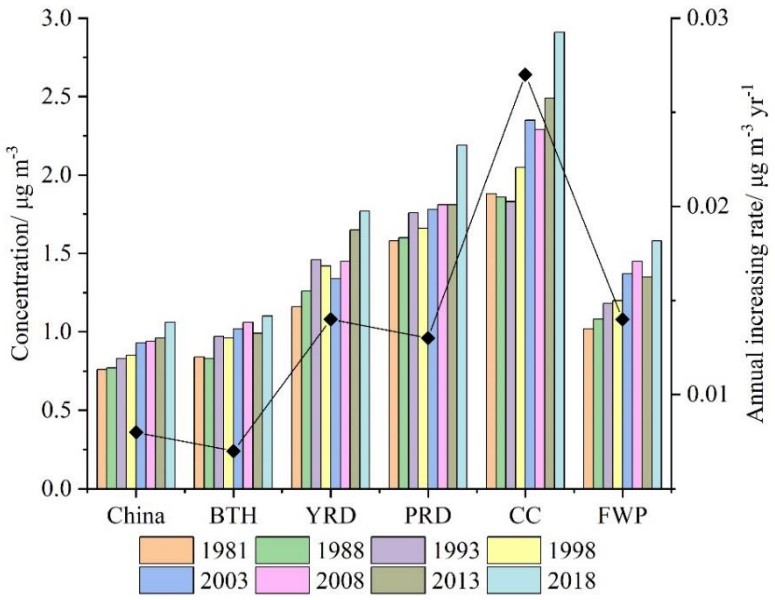

**Fig. 8.** The interannual changes of SOA in China and the key regions.

## 4 Conclusions

BVOC emissions play a key role in the formation of O$_3$ and SOA. In the summer of 2018, China's BVOC emissions of 9.91 Tg cause ambient O$_3$ and SOA concentrations to increase by 8.6 ppb (16.75%) and 0.84 μg m$^{-3}$ (73.15%) on average,

respectively. The impacts of BVOCs have an obvious spatial difference that $O_3$ and SOA in most southern regions show high sensitivities to BVOCs. Due to the different emissions and level of biogenic and anthropogenic precursors, BVOC emissions have distinguished impacts in different regions. CC region has the highest $O_3$ and SOA generated by BVOCs. Some areas with equivalently high BVOC emissions however have different contributions to $O_3$, such as northeastern and southern China. The impacts of BVOCs are affected not only by the relative abundance of biogenic and anthropogenic VOCs but also by the VOCs/$NO_x$ ratio. The sensitivity to BVOCs differs over regions. To decrease BVOC emissions by planting plants with low emission potential may contribute to $O_3$ pollution control in most regions of southern, central, and northeastern China. For the abatement of BSOA in summer, the decreased plantation of needleleaf trees or the replacement by trees with low monoterpene emission potential are expected to be helpful.

Considering the increasing vegetation coverage and greening trend in China in recent decades (Chen et al., 2019; Piao et al., 2015), the resulted changes in leaf biomass will influence BVOC emissions, which then affect the formation of $O_3$ and SOA. The interannual variation of BVOC emissions caused by increasing leaf biomass results in significant increases (p<0.05) of $O_3$ and SOA concentrations at average rates of 0.11 ppb yr$^{-1}$ and 0.008 μg m$^{-3}$ yr$^{-1}$, respectively. It shows different interannual variations which can be attributed to the differences in changing trends of leaf biomass. The southern region with obvious increase of leaf biomass showed large enhanced $O_3$ and SOA. In the future, in order to achieve the goal of carbon neutrality, China will not only reduce carbon emissions through energy conservation and emission reduction, but also increase carbon sinks through the development of carbon sequestration technologies and biological carbon sinks (Wang and Zhang, 2020). Increasing forest carbon sinks will inevitably lead to an increase in vegetation coverage, so that BVOC emissions will continue to increase, leading to higher contributions to $O_3$ and SOA production in the future. Therefore, studying the influence of land cover changes on BVOC emissions and their impact on the generation of $O_3$ and SOA is of great significance for future researches on precise prevention and control of air pollution in China in the context of fighting climate change.

In this study, we aim to explore the impact of interannual BVOC emission variations on $O_3$ and SOA caused by vegetation biomass variability. The historical $O_3$ and SOA concentrations were simulated by fixing the anthropogenic emissions and meteorological data in 2008. The influences of annual meteorology on BVOC emissions and formation of secondary air pollutants were not considered. Although vegetation change is the main driver of interannual variations of BVOC emissions, it still brings uncertainty to the simulation in this study. Future work can update the anthropogenic emission inventory and use dynamic meteorological data to explore multivariate effects and provide more accurate data for evaluating the roles of biogenic emissions in air quality. At the same time, the impact of meteorological changes on the long-term changes of BVOC emissions and formation of secondary air pollutants should also be considered to provide a scientific basis for the precise prevention and control of air pollution in response to climate change.

**Code and data availability**

The code or data used in this study are available upon request from the corresponding author Lingyu Li (lilingyu@qdu.edu.cn).

**Author contribution**

LL planned and organized the project. SS provided the code, JC performed the simulation, analysis and writing, YH made adjustments of the paper's visualization. SX, YH, LL revised the manuscript.

**Competing interests**

The authors declare that they have no conflict of interest.

**Acknowledgments**

This work was supported by National Natural Science Foundation of China (42075103, 41705098) and Science and Technology Support Plan for Youth Innovation of Colleges in Shandong Province (DC2000000961).

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
