# Peer review of "Enhanced summertime ozone and SOA from biogenic volatile organic compound (BVOC) emissions due to vegetation biomass variability during 1981–2018 in China"

_Atmospheric Chemistry and Physics, 2021_

## Author Comment (AC1)

**Response to Referee #1**

*1. Model evaluation should be added. The authors should show evidences to make the results convincing, especially the BVOC emissions.*

**Response:** Thank you so much for your suggestion. In revised manuscript, we add the model evaluations.

(1) Validation of BVOC emission simulations by MEGAN.

Firstly, we compared the estimation with canopy-level emission flux measurements in China (Bai et al., 2015, 2016, 2017). The gridded BVOC emission estimated by MEGAN were extracted where locating the flux measurement sites. The comparison of model simulation and observation is shown in Fig. S1. The estimated BVOC emissions are higher with an average mean bias of 1.11 mg m$^{-2}$ h$^{-1}$, mainly because of the differences in time between the simulation and measurements. But they are correlated by r=0.84, exhibiting good agreement in spatial variations.

Secondly, formaldehyde (HCHO) concentrations observed by satellite were used to evaluate the spatial variability of estimated isoprene emissions. Isoprene is the dominant compound among BVOC species that accounts for almost half of total BVOC emissions in China. Since HCHO is an important proxy of isoprene in forest regions with no significant anthropogenic impact, satellite HCHO observations can be used to validate the spatial variability of biogenic isoprene emissions. We compared simulated isoprene emission with satellite-derived HCHO column concentration using the Ozone Monitoring Instrument (OMI) HCHO vertical column product. The monthly averaged OMI HCHO vertical column in June 2018 correlates with the model estimated results at a 99% confidence level.

Line 171–185 of revised manuscript, "The emission simulations were validated by using the measurements of BVOC emission flux and formaldehyde (HCHO) concentration. The flux measurements of BVOCs conducted in China were collected (Bai et al., 2015, 2016, 2017). The gridded BVOC emission estimated by MEGAN were extracted where the flux measurement sites were located to do the comparison (Fig. S1). The modeled fluxes of BVOCs in this study capture the spatial variability of observations better with a correlation coefficient of 0.84. But

the estimation is higher than measurement with an average mean bias of 1.11 mg m$^{-2}$ h$^{-1}$, mainly because of the differences in time between them. Isoprene is the main compound in BVOC species, accounting for nearly half of total BVOC emissions in China. It undergoes chemical and photochemical reactions in the atmosphere, and the oxidation product is mainly HCHO (Bai and Hao, 2018; Orlando et al., 2000). In forest areas and in summer, biogenic isoprene is the dominant source of HCHO, so satellite HCHO column concentration is widely used to constrain isoprene emissions (Opacka et al., 2021; Palmer et al., 2003; Stavrakou et al., 2018; Wang et al., 2021; Zhang et al., 2021). In this study, we used the HCHO vertical column detected by Ozone Monitoring Instrument (OMI) to validate the spatial variability of isoprene estimates. The monthly OMI HCHO data from the EU FP7 project QA4ECV product (Quality Assurance for Essential Climate Variables; http://www.qa4ecv.eu) was used in this study. The result of the statistical analysis with a confidence interval of 99% indicates that the monthly averaged OMI HCHO vertical column in June 2018 is significantly correlated to the model-estimated isoprene emissions." is added.

[Figure]

**Fig. S1.** Comparison of MEGAN model simulations with flux measurements in China.

(2) Validation of meteorological data simulated by WRF.

Temperature and radiation play key roles in BVOC emissions. The observations of temperature at 2411 sites in 2008 and 684 sites in 2018 in China were used to compare with

WRF-simulated 2-m temperature (T2). The simulated radiation was not evaluated because of a lack of available site observations. In the revised manuscript, we add the validation for WRF simulation. Meanwhile, to make the description more clear, the statement of MEGAN in Section 2.1 is reorganized.

Line 87–96, "Meteorology, gridded fraction of plant functional types (PFTs), PFT-specific emission factors, and leaf area index (LAI) are inputs to drive MEGAN. MODIS LAI data was used. The hourly meteorological fields including temperature, downward shortwave radiation, wind speed, water vapor mixing ratio, pressure, and precipitation were simulated by the WRF model in this study. The WRF-simulated meteorological fields were verified to be considered reasonable for driving MEGAN (Li et al., 2013, 2021)." is revised to "MEGAN2.1 requires hourly weather variables to drive the calculation of hourly BVOC emissions. The hourly meteorological fields including temperature, downward shortwave radiation, wind speed, water vapor mixing ratio, pressure, and precipitation were simulated by the WRF model in this study. Temperature and radiation play key roles in BVOC emissions. We used the observed daily average temperature at 2411 sites in 2008 and 684 sites in 2018 in China to evaluate the reliability of the 2-m temperature (T2) simulated by WRF in this study. The observations were from the National Meteorological Data Center in China (http://data.cma.cn/). The simulated radiation was not evaluated because of a lack of available site observations. For 2008, the average mean bias (MB), mean absolute error (MAE), and root-mean-square error (RMSE) are 0.36, 2.47, and 3.30 K over China. For 2018, these statistics are 1.24, 2.46 and 3.30 K, respectively. The correlation coefficients between simulations and observations are 0.82 and 0.86 for the year 2008 and 2018, respectively. In general, the WRF simulation is considered reasonable for driving MEGAN.". Line 87, "Meteorological and vegetation data are inputs to drive MEGAN." is added. Line 96–98, "The vegetation data includes gridded fraction of plant functional types (PFTs), leaf area index (LAI), and PFT-specific emission factors." is added. Line 103, "For LAI, the MODIS LAI data was used." is added.

(3) Validation of $O_3$ and SOA simulations by WRF-Chem.

In this study, we aim to explore the impact of interannual BVOC emission variations on $O_3$ and SOA caused by vegetation biomass variability. The historical $O_3$ and SOA concentrations were simulated by fixing the anthropogenic emissions and meteorological data

in 2008. So we did not conduct validation for WRF-Chem simulation. However, WRF-Chem model has been widely used in global and regional pollution studies and verified to have a good performance on the simulation of secondary pollutants(Gupta and Mohan, 2015; Hoshyaripour et al., 2016; Li et al., 2018; Situ et al., 2013; Wu et al., 2018).

Line 129–131 of revised manuscript, "A large number of global and regional air pollution studies widely apply it to simulate secondary pollutants, and the verification results show that it can well reproduce the observed pollutant concentrations (Gupta and Mohan, 2015; Hoshyaripour et al., 2016; Li et al., 2018; Situ et al., 2013; Wu et al., 2018)." is added.

In addition, the uncertainty by fixing the anthropogenic emissions and meteorological data in 2008 is also added in the revised manuscript. Line 402–410, "In this study, we aim to explore the impact of interannual BVOC emission variations on $O_3$ and SOA caused by vegetation biomass variability. The historical $O_3$ and SOA concentrations were simulated by fixing the anthropogenic emissions and meteorological data in 2008. The influences of annual meteorology on BVOC emissions and formation of secondary air pollutants were not considered. Although vegetation change is the main driver of interannual variations of BVOC emissions, it still brings uncertainty to the simulation in this study. Future work can update the anthropogenic emission inventory and use dynamic meteorological data to explore multivariate effects and provide more accurate data for evaluating the roles of biogenic emissions in air quality. At the same time, the impact of meteorological changes on the long-term changes of BVOC emissions and formation of secondary air pollutants should also be considered to provide a scientific basis for the precise prevention and control of air pollution in response to climate change." is added.

References:

[revised manuscript text omitted]

*2. The manuscript focused on the change of background $O_3$ and SOA in China due to the change of vegetation The MS should show the change of biomass as well as how the biomass changes affecting BVOC emissions.*

**Response:** Thank you very much for your valuable suggestion. In revised manuscript, we add a new Section of "**3.1 BVOC emission**", in which "**3.1.2 Influence of leaf biomass**

**variability**" describes the interannual variation of leaf biomass and its influence on BVOC emissions. The added Section 3.1.2 is as follows.

**"3.1.2 Influence of leaf biomass variability**

The leaf biomasses increased from $378.35 \times 10^{12}$ g in 1981 to $1107.16 \times 10^{12}$ g in 2018 at an average rate of $17.97 \times 10^{12}$ g $yr^{-1}$. Among them, the forest and crop leaf biomass increased from $237.10 \times 10^{12}$ to $518.38 \times 10^{12}$ g and from $141.25 \times 10^{12}$ to $588.79 \times 10^{12}$ g, respectively, totally increasing by 192.63%. The spatial distribution of interannual variations in leaf biomass is presented in Fig. S2. The increase of leaf biomass is most significant in Great Khingan, Changbai Mountains, North China Plain, south and southwest China. This is mainly due to the increased stock of broadleaf and coniferous forests as a result of afforestation. Northern Qinghai-Tibet area and Northwest China have a relatively high grass cover rate but insignificant increase in leaf biomass of vegetation. It is because that the grass biomasses were the same over the historical simulations due to lacking of data.

Due to the increased volume and production of vegetation, the total BVOC emissions increased by 58.66% at average rates of 96.64 Gg $yr^{-1}$, of which isoprene, monoterpene, sesquiterpene increased by 108.57%, 38.17%, and 33.35% at average rates of 11.10, 0.99, and 0.17 Gg $yr^{-1}$, respectively. Isoprene emissions increased more rapidly over the past 40 years, which is primarily due to the greater increase in the biomass of broadleaf trees, which have the highest isoprene emission rates. Monoterpene and sesquiterpene increased at a lower rate because the increase of leaf biomass of conifers is relatively small. Fig. 2 shows the spatial distribution of interannual variations in BVOC emissions caused by the changing leaf biomass. Since the needleleaf and broadleaf trees tend to have a higher emission potential than grass or crop (Guenther et al., 2012), their wide distribution and the substantial increase in biomass result in the largest interannual variability of BVOC emissions in the Great Khingan, Changbai Mountains, North China Plain, Central and Southern China, and Hainan Province. However, the emission of BVOCs in the northwest and southern coastal areas has decreased.

[Figure]

**Fig. 2.** Spatial distribution of interannual variations in BVOC emissions caused by leaf biomass changes."

Line 310–314, "The change of vegetation leaf biomass will cause the interannual variation of BVOCs emissions and then $O_3$ and SOA generation. During 1981–2018, the forest and crop leaf biomass increased from $237.10 \times 10^{12}$ g to $518.38 \times 10^{12}$ g and from $141.25 \times 10^{12}$ g to $588.79 \times 10^{12}$ g, respectively, totally increasing by 192.63%. In this study, the annual emission factors extrapolated from emission rates and annual leaf biomass were used to simulate the impact of interannual variation in BVOC emissions on $O_3$ and SOA, as described in Scenario "HISTORY"." is deleted.

Figure S2 is added to the Supplementary File.

[Figure]

**Fig. S2.** Spatial distribution of interannual variations in leaf biomass.

**Table S1.** WRF-Chem configuration.

| Parameter | Option |
|---|---|
| Microphysics | Purdue Lin et al. scheme |
| Long-wave radiation | RRTM scheme |
| Short-wave radiation | Goddard shortwave scheme |
| Surface layer | Monin-Obukhov theory |
| Land surface | Noah Land Surface Model |
| Cumulus parameter | Grell-Devenyi Ensemble scheme |
| Planetary Boundary Layer | YSU scheme |
| Gas phase chemistry scheme | NOAA/ESRL RACM |
| Aerosol chemistry scheme | VBS |
| Photolysis scheme | Fast-J |

*7. Table 2: Don't branch the percentages of BVOC contribution*

**Response:** Thank you for your suggestion. The percentages are deleted in the revised Table 2, as follows.

**Table 2.** Emissions of each BVOC category and their corresponding contribution to MDA8 $O_3$ and SOA concentration in the five key regions of China in June 2018.

| | BVOC category | China | BTH | FWP | PRD | YRD | CC |
|---|---|---|---|---|---|---|---|
| Emission ($10^4$ tons) | Isoprene | 636.28 | 15.66 | 30.29 | 34.74 | 53.43 | 24.50 |
| | Monoterpene | 104.81 | 2.77 | 2.16 | 5.09 | 4.38 | 11.59 |
| | Sesquiterpene | 20.98 | 0.64 | 0.32 | 1.27 | 1.42 | 1.65 |
| | Total BVOCs | 990.91 | 29.23 | 40.67 | 46.29 | 74.18 | 51.32 |
| Contribution to MDA8 $O_3$ (ppb) | Isoprene | 7.01 | 3.42 | 18.01 | 16.81 | 12.55 | 20.49 |
| | Monoterpene | 1.17 | 1.74 | 0.32 | 4.07 | 2.92 | 6.89 |
| | Sesquiterpene | 0.16 | 0.93 | -1.36 | 1.66 | 1.06 | 2.88 |
| | Isoprenoid | 7.77 | 2.94 | 17.43 | 16.34 | 14.23 | 24.55 |
| | Total BVOCs | 8.61 | 4.10 | 18.94 | 18.74 | 13.40 | 23.29 |
| Contribution to SOA ($\mu g\ m^{-3}$) | Isoprene | 0.25 | 0.20 | 0.53 | 0.95 | 0.63 | 0.91 |
| | Monoterpene | 0.52 | 0.45 | 0.72 | 1.21 | 0.62 | 1.59 |
| | Sesquiterpene | 0.22 | 0.21 | 0.26 | 0.49 | 0.31 | 0.75 |
| | Isoprenoid | 0.84 | 0.78 | 1.30 | 1.96 | 1.29 | 2.52 |
| | Total BVOCs | 0.84 | 0.74 | 1.29 | 1.96 | 1.27 | 2.51 |

*8. The form of picture display should be strengthened*

**Response:** Thank you for your suggestion. In the revised manuscript, Fig. 4–8 are revised to make them more clear. Firstly, in Fig. 4, 5, and 7, the five key regions are marked. Secondly, Fig. 6 and 8 are changed by adding a coordinate of the annual increasing rate to better display the differences in the five key regions.

[Figure]

**Fig. 4.** Spatial variations in impact of BVOC emission on SOA concentration.

[Figure]

**Fig. 5.** Spatial distribution of interannual variations in $O_3$ simulated using annual BVOC emission factors.

[Figure]

**Fig. 6.** The interannual changes of O₃ in China and the key regions.

[Figure]

**Fig. 7.** Spatial distribution of interannual variations in SOA simulated using annual BVOC emission factors.

[Figure]

**Fig. 8.** The interannual changes of SOA in China and the key regions.

---

## Author Comment (AC2)

**Response to Referee #2**

*1. Section 2.1: I recommend that you show a spatial map of changes in BVOCs over time? Li et al. (2020) shows only 2008-2018.*

**Response:** Thank you for your valuable suggestion. In this study, the historical BVOC emissions were estimated using the same meteorology data to explore the influence of the interannual variability of vegetation biomass. So we add a spatial map of changes in BVOC emissions caused by vegetation biomass variability, as shown in Fig. 2. In addition, we add a new Section 3.1 "BVOC emissions" to describe the BVOC emissions in June 2018 and their interannual changes. The relative description of BVOC emissions in Section 2.1 of the original manuscript are moved here. In Section 3.1.2 "Sensitivity of the BVOC emissions to leaf biomass", Figure 2 is added and the changes in BVOC emissions caused by vegetation biomass variability are discussed.

Line 195–204 in revised manuscript, "Due to the increased volume and production of vegetation, the total BVOC emissions increased by 58.66% at average rates of 96.64 Gg yr$^{-1}$, of which isoprene, monoterpene, sesquiterpene increased by 108.57%, 38.17%, and 33.35% at average rates of 11.10, 0.99, and 0.17 Gg yr$^{-1}$, respectively. Isoprene emissions increased more rapidly over the past 40 years, which is primarily due to the greater increase in the biomass of broadleaf trees, which have the highest isoprene emission rates. Monoterpene and sesquiterpene increased at a lower rate because the increase of leaf biomass of conifers is relatively small. Fig. 2 shows the spatial distribution of interannual variations in BVOC emissions caused by the changing leaf biomass. Since the needleleaf and broadleaf trees tend to have a higher emission potential than grass or crop (Guenther et al., 2012), their wide distribution and the substantial increase in biomass result in the largest interannual variability of BVOC emissions in the Great Khingan, Changbai Mountains, North China Plain, Central and Southern China, and Hainan Province. However, the emission of BVOCs in the northwest and southern coastal areas has decreased."

[Figure]

**Fig. 2.** Spatial distribution of interannual variations in BVOC emissions caused by leaf biomass changes.

References:

Guenther, A. B., Jiang, X., Heald, C. L., Sakulyanontvittaya, T., Duhl, T., Emmons, L. K., and Wang, X.: The model of emissions of gases and aerosols from nature version 2.1 (MEGAN2.1): an extended and updated framework for modeling biogenic emissions, Geosci. Model Dev., 5, 1471–1492, https://doi.org/10.5194/gmd-5-1471-2012, 2012.

*2. In the introduction your say that your study uses "more accurate BVOC emissions". However, you don't provide evidence of this. Satellite data of formaldehyde could help in the evaluation over the satellite era. Do a literature search as there are a number of studies that use HCHO data to constrain BVOC emissions. How will you validate your emissions before the satellite era (e.g., 1981–2004)?*

**Response:** Thank you for your constructive comment and we are sorry for the unclear explanation for our "more accurate BVOC emissions".

**Firstly**, we outstand the higher accuracy through comparing the method between our study and others in BVOC emission estimation. (1) **Determination of emission rate and leaf biomass.** In previous studies, traditional emission categories were used to determine isoprene

and monoterpene emission rates (Guenther et al., 1994; Klinger et al., 2002; Simpson et al., 1999; Wang et al., 2007). In this method, discrete emission categories (e.g., negligible, low, moderate, and high) were defined, which lacked theoretical evidence. And some studies utilized coarse emission intensity classifications. The determined emission rates typically differed between studies and had high uncertainty. In our study, we summarized a large number of observations from China and other countries to obtain more accurate basal emission rates by the theoretically effective statistical approach (Li et al., 2020). The statistical isoprene emission rates included seven categories by lowest, lower, low, moderate, high, higher, and highest. Monoterpene included six categories by lowest, lower, low, moderate, high, and higher. The accuracy of emission rates can be expected to be improved. For leaf biomass, the previous studies usually applied an average value for each vegetation class, such as broadleaf trees, needleleaf trees, crops, and grasses, without revealing their differences among regions and plant species (Klinger et al., 2002; Wang et al., 2007). In our study, the plant specific leaf biomasses were estimated based on the provincial or city-level statistic of vegetation volume and production using apportion models (Li and Xie, 2014). We obtained the gridded leaf biomass for the 23 vegetation species/types. (2) **Detailed vegetation classification.** Most studies on the BVOC emissions inventory in China typically include a coarse vegetation classification that is based on a less-detailed vegetation distribution (Gao et al., 2019; Klinger et al., 2002; Wang et al., 2007). And the MEGAN2.1 defined 15 vegetation types (Guenther et al., 2012). In our study, Vegetation Atlas of China (1:1,000,000), which had detailed vegetation distributions at a high horizontal resolution of about 250 m, was used to produce more detailed vegetation classification in Shandong Province, including 23 plant species/types (four broadleaf trees, five needleleaf trees, eight crop species, and six subtypes of shrub and grass).

In the revised manuscript, we add the explanation to show the higher accuracy of the determined emission factors and vegetation classification. Line 101–103, "Previous studies typically included a coarse vegetation classification that is based on a less-detailed vegetation distribution (Gao et al., 2019; Klinger et al., 2002; Wang et al., 2007). And the MEGAN2.1 defined 15 vegetation types by default (Guenther et al., 2012)." is added. Line 109–111, "In previous studies, traditional emission categories were used to determine emission rates (Guenther et al., 1994; Klinger et al., 2002; Simpson et al., 1999; Wang et al., 2007), which

usually utilized coarse categories and resulted in high uncertainty." is added. Line 113–115, "Previous studies usually applied an average value for each vegetation class, such as broadleaf trees, needleleaf trees, crops, and grasses, without revealing their differences among regions and plant species (Klinger et al., 2002; Wang et al., 2007)." is added.

**Secondly**, as you suggested, we add the validation of BVOC emissions by comparing the simulation with satellite observation of formaldehyde concentration and also the canopy-level emission flux measurements in China. The validation is conducted for estimation in 2018. The emissions in other years are not evaluated because the historical BVOC emissions are estimated using the same meteorology data to explore the influence of the interannual variability of vegetation biomass. They cannot display the real emissions for the years other than 2018. (1) Validation by comparing with canopy-level emission flux measurements. The canopy-level emission flux measurements in China were used for validation (Bai et al., 2015, 2016, 2017). The gridded BVOC emission estimated by MEGAN were extracted where locating the flux measurement sites. The comparison of model simulation and observation is shown in Fig. S1. The estimated of BVOC emissions are higher with an average mean bias of 1.11 mg m$^{-2}$ h$^{-1}$, mainly because of the differences in time between the simulation and measurements. But they are correlated by r=0.84, exhibiting good agreement in spatial variations. (2) Validation by comparing with satellite observation of formaldehyde concentration. We compared simulated isoprene emission with satellite-derived HCHO column concentration using the Ozone Monitoring Instrument (OMI) HCHO vertical column product. The monthly averaged OMI HCHO vertical column in June 2018 correlates with the model estimated results at a 99% confidence level.

[revised manuscript text omitted]

(v3.6) and its impacts over eastern China, Geosci. Model Dev., 14, 6155–6175,

https://doi.org/10.5194/gmd-14-6155-2021, 2021.

*3. Table 1: Why do you use "Year 2008" for your "HISTORY" run, but "Year 2018" for the rest of your simulations?*

**Response:** In the "HISTORY" simulation, we aimed to explore the impacts of interannual BVOC emission variations on $O_3$ and SOA formation caused by vegetation biomass variability during 1981–2018. In order to achieve this goal, we designed a scenario experiment by fixing meteorological data in one year and using the annual leaf biomass to drive BVOC estimates. The influences of annual meteorology on BVOC emissions and formation of secondary air pollutants were not considered. It is reported that vegetation change is the main driver of interannual variations of BVOC emissions (Li et al., 2020; Wang et al., 2021). The large-scale afforestation activities in recent years lead to the rapid increase of vegetation leaf biomass and therein BVOC emissions. In addition, the annual average temperature has a heating rate of 0.26 K per 10 years from 1951 to 2020 (Climate Change Center of China Meteorological Administration, 2021). Taking into account the long-term warming trend, we chose meteorological data of a mid-year 2008 over 1981–2018 as the constant input for historical simulation. The scenarios "BASE", "BIO", "ISOP", "MTP", "SQT", and "ISOPRENOID" were designed to simulate the impacts of emissions of BVOCs, isoprene, monoterpenes, sesquiterpenes, and isoprenoids (total of isoprene, monoterpene and sesquiterpene emissions) in June 2018 on $O_3$ and SOA, respectively. So we used the meteorology of 2018 to drive MEGAN to estimate the emissions in 2018.

We add the explanation in the revised manuscript. Line 156–157, "For these simulations, the meteorology of 2018 was used to drive MEGAN to estimate biogenic emissions in June 2018." is added. Line 159–162, "For the meteorology, the fixing set of a mid-year 2008 over 1981–2018 were used for all the HISTORY simulations. To explore the impacts of interannual BVOC emission variations caused by vegetation biomass variability, influences of annual

meteorology on BVOC emissions and formation of secondary air pollutants were not considered." is added.

**Response:** Thank you for your valuable suggestion. In the revised manuscript, the five key regions are marked in Figures 4, 5, and 7.

[Figure]

**Fig. 4.** Spatial variations in impact of BVOC emission on SOA concentration.

[Figure]

**Fig. 5.** Spatial distribution of interannual variations in $O_3$ simulated using annual BVOC emission factors.

[Figure]

**Fig. 7.** Spatial distribution of interannual variations in SOA simulated using annual BVOC emission factors.

*6. Line 245: "N3"? Do you mean NO3?*

**Response:** We appreciate your careful reading very much and are sorry for the error. In revised manuscript, line 288, it is revised to "$NO_3$".

*7. Line 275: You don't show the BVOCs emission changes for your study in "HISTORY". Why? A spatial map of the changes would be very helpful for the discussion.*

**Response:** Thank you for your suggestion. In the revised manuscript, we add the detailed description of BVOC emission changes for the "HISTORY" simulation in **Section3.1.2**. The spatial map of the changes is also added as Fig. 2. In addition, the reason for the changes is discussed. The revisions can refer to the **Response to Comment 1 of Reviewer 2**.

*8. Section 3.3.1: The discussion would be facilitated by maps (e.g., in supplementary material) of vegetation changes, leaf biomass changes, emission factors, etc.*

**Response:** Thank you for your suggestion. In the revised manuscript, we add a spatial map of leaf biomass changes over 1981–2018 as Fig. S2. The vegetation distribution in our

estimations was derived from the Vegetation Atlas of China (1:1,000,000), which provides a detailed vegetation distribution at a high resolution. It was produced based on nationwide vegetation surveys and research from the past four decades and was published in 2007. It is rarely updated, and it represents the average distribution of vegetation in China. Therefore, the same PFT distribution was used to estimate BVOC emissions for 1981–2018 in this study. Emission factors were extrapolated by the leaf-level emission rates and leaf biomass using the canopy environment model described in MGEAN2.1. Emission rates were constant by time. So the interannual change of emission factor can be explained by that of leaf biomass.

Line 187–194 in revised manuscript, we add descriptions for the changes of leaf biomass: "The leaf biomasses increased from $378.35 \times 10^{12}$ g in 1981 to $1107.16 \times 10^{12}$ g in 2018 at an average rate of $17.97 \times 10^{12}$ g yr$^{-1}$. Among them, the forest and crop leaf biomass increased from $237.10 \times 10^{12}$ to $518.38 \times 10^{12}$ g and from $141.25 \times 10^{12}$ to $588.79 \times 10^{12}$ g, respectively, totally increasing by 192.63%. The spatial distribution of interannual variations in leaf biomass is presented in Fig. S2. The increase of leaf biomass is most significant in Great Khingan, Changbai Mountains, North China Plain, south and southwest China. This is mainly due to the increased stock of broadleaf and coniferous forests as a result of afforestation. Northern Qinghai-Tibet area and Northwest China have a relatively high grass cover rate but insignificant increase in leaf biomass of vegetation. It is because that the grass biomasses were the same over the historical simulations due to lacking of data.".

[Figure]

**Fig. S2.** Spatial distribution of interannual variations in leaf biomass.

---

## Author Response (AR2)

**Author's response**

Manuscript: "Enhanced summertime ozone and SOA from biogenic volatile organic compound (BVOC) emissions due to vegetation biomass variability during 1981–2018 in China" (acp-2021-675)

Responses to reviewers' comments are listed below, respectively. Revised portions are marked in red.

**Response to Referee #1**

*1.The validation for WRF/Chem simulation hasn't been conducted. Even the WRF/Chem model is a very powerful and advanced model for simulating regional atmospheric environment, simulation errors are always present. The MS should show how much error in the base to help readers to get how real the model results are. Evaluations for ozone and $PM_{2.5}$ in the base run are necessary, in my opinion.*

**Response:** Thank you so much for your suggestion. In the revised manuscript, we add the validation of air quality simulations by WRF-Chem.

In this study, we focused on the contribution of historical BVOC emissions caused by vegetation biomass variability, so the anthropogenic emissions in all scenarios were fixed using the MIX Asian anthropogenic source emission inventory. In the base run, both anthropogenic and BVOC emissions were inputted to conduct $O_3$ and SOA simulation in June 2018. The observations of daily maximum 8-h (MDA8) $O_3$ and daily average $PM_{2.5}$ at 1588 sites in China were applied to evaluate the WRF-Chem simulation in the base run.

Line 143–153 of revised manuscript, "The observed daily maximum 8-h (MDA8) $O_3$ and daily average $PM_{2.5}$ concentrations at 1588 sites in June 2018 in China were applied to evaluate the WRF-Chem simulations in the control run (as listed in Table 1) in this study. The observations were from the daily updated national air quality released by the China National Environmental Monitoring Centre (http://www.cnemc.cn/). The verification statistics are

shown in Table S1. Notably, PM$_{2.5}$ had no systematic bias between the observation and simulation, while the model predicted O$_3$ concentrations were lower than measurements. The errors could be mainly attributed to the anthropogenic emissions data used in this study as described in Section 2.3." is added.

**Table S1.** Verification statistics of meteorology and air quality simulations.

| Variable | Year | Mean | | MB | MAE | RMSE |
| --- | --- | --- | --- | --- | --- | --- |
| | | Observation | Simulation | | | |
| T2 (K) | 2008 | 295.84 | 295.48 | 0.36 | 2.47 | 3.30 |
| | 2018 | 295.83 | 294.60 | 1.24 | 2.46 | 3.30 |
| MDA8 O$_3$ (ppb) | 2018 | 58.83 | 36.39 | 22.44 | 36.65 | 45.42 |
| PM$_{2.5}$ (µg m$^{-3}$) | 2018 | 29.29 | 29.81 | -2.71 | 21.31 | 30.51 |

T2: temperature at 2 m; MB: mean bias; MAE: mean absolute error; RMSE: root mean square error.

*2. The MS considers REA results to validate the MEGAN performance, which maybe the most direct verification for BVOCs emissions. It is good. But can the MS supply the information of sampling location? And the conclusion that "the estimation is higher than measurement with an average mean bias of 1.11 mg m$^{-2}$ h$^{-1}$" should be noted that it estimated higher with a mean bias of 1.1.1 mg/m$^2$/hr in these REA sampling regions, but not for the whole China in averaged. And it should be better to show the comparison between HCHO and MEGAN results first, followed by the comparison between REA and MEGAN results.*

**Response:** Thank you very much for your valuable suggestion. In revised manuscript, we have added specific information on sampling sites and provided a clearer explanation about the use of flux measurement to evaluate BVOCs simulations limited to these sampling regions. Additionally, we adjusted the order of the flux measurements evaluation and HCHO concentration evaluation.

Line 186–203, "The emission simulations were validated by using the measurements of BVOC emission flux and formaldehyde (HCHO) concentration. The flux measurements of

BVOCs conducted in China were collected (Bai et al., 2015, 2016, 2017). The gridded BVOC emission estimated by MEGAN were extracted where the flux measurement sites were located to do the comparison (Fig. S1). The modeled fluxes of BVOCs in this study capture the spatial variability of observations better with a correlation coefficient of 0.84. But the estimation is higher than measurement with an average mean bias of 1.11 mg m$^{-2}$ h$^{-1}$, mainly because of the differences in time between them. Isoprene is the main compound in BVOC species, accounting for nearly half of total BVOC emissions in China. It undergoes chemical and photochemical reactions in the atmosphere, and the oxidation product is mainly HCHO (Bai and Hao, 2018; Orlando et al., 2000). In forest areas and in summer, biogenic isoprene is the dominant source of HCHO, so satellite HCHO column concentration is widely used to constrain isoprene emissions (Opacka et al., 2021; Palmer et al., 2003; Stavrakou et al., 2018; Wang et al., 2021; Zhang et al., 2021). In this study, we used the HCHO vertical column detected by Ozone Monitoring Instrument (OMI) to validate the spatial variability of isoprene estimates. The monthly OMI HCHO data from the EU FP7 project QA4ECV product (Quality Assurance for Essential Climate Variables; http://www.qa4ecv.eu) was used in this study. The result of the statistical analysis with a confidence interval of 99% indicates that the monthly averaged OMI HCHO vertical column in June 2018 is significantly correlated to the model-estimated isoprene emissions." is revised to "The emission simulations were validated by using the formaldehyde (HCHO) concentration and measurements of BVOC emission flux. Isoprene is the main compound in BVOC species, accounting for nearly half of total BVOC emissions in China. It undergoes chemical and photochemical reactions in the atmosphere, and the oxidation product is mainly HCHO (Bai and Hao, 2018; Orlando et al., 2000). In forest areas and in summer, biogenic isoprene is the dominant source of HCHO, so satellite HCHO column concentration is widely used to constrain isoprene emissions (Opacka et al., 2021; Palmer et al., 2003; Stavrakou et al., 2018; Wang et al., 2021; Zhang et al., 2021). In this study, we used the HCHO vertical column detected by Ozone Monitoring Instrument (OMI) to validate the spatial variability of isoprene estimates. The monthly OMI HCHO data from the EU FP7 project QA4ECV product (Quality Assurance for Essential Climate Variables; http://www.qa4ecv.eu) was used in this study. The result of the statistical analysis with a confidence interval of 99% indicates that the monthly averaged OMI HCHO vertical column in June 2018 is significantly

correlated to the model-estimated isoprene emissions. The flux measurements of BVOCs conducted in China were collected (Bai et al., 2015, 2016, 2017). The details of these measurements are provided in Table S3, including the location of sampling sites and observed BVOC emission fluxes. The gridded BVOC emission estimated by MEGAN were extracted where the flux measurement sites were located to do the comparison (Fig. S1). The modeled fluxes of BVOCs in this study capture the spatial variability of observations better with a correlation coefficient of 0.84. But the estimation is higher than measurement with an average mean bias of 1.11 mg m$^{-2}$ h$^{-1}$ in these sampling sites, mainly because of the differences in time between them.".

**Table S3.** Detailed descriptions of the flux measurements used in this study.

| Site | Location | Isoprene | | Monoterpene | | Sesquiterpene | | References |
|------|----------|------|------|------|------|------|------|------------|
| | | Obs | Sim | Obs | Sim | Obs | Sim | |
| Qianyanzhou | 26°44′48″N, 115°04′13″E | 0.07 | 3.25 | 0.81 | 0.17 | 0.01 | 0.06 | Bai et al. (2017) |
| Taihuyuan | 30°18′N, 119°34′E | 3.35 | 7.06 | 0.01 | 0.14 | 0.00 | 0.02 | Bai et al. (2016) |
| Changbai Mountain | 42°240′N, 128°60′E | 0.95 | 4.03 | 0.14 | 0.72 | 0.19 | 0.05 | Bai et al. (2015) |

Obs: observation; Sim: simulation; the units of flux measurements are mg m$^{-2}$ h$^{-1}$.

References:

Bai, J. H. and Hao, N.: The relationships between biogenic volatile organic compound (BVOC) emissions and atmospheric formaldehyde in a subtropical Pinus plantation in China, Ecology and Environmental Sciences, 27(6): 991–999, https://doi.org/10.16258/j.cnki.1674-5906.2018.06.001, 2018.

Bai, J., Guenther, A., Turnipseed, A., and Duhl, T.: Seasonal and interannual variations in whole-ecosystem isoprene and monoterpene emissions from a temperate mixed forest in Northern China, Atmos. Pollut. Res., 6, 696–707, https://doi.org/10.5094/APR.2015.078, 2015.

Bai, J., Guenther, A., Turnipseed, A., Duhl, T., Yu, S., and Wang, B.: Seasonal variations in whole-ecosystem BVOC emissions from a subtropical bamboo plantation in China, Atmos. Environ., 124, 12–21, https://doi.org/10.1016/j.atmosenv.2015.11.008, 2016.

Bai, J., Guenther, A., Turnipseed, A., Duhl, T., and Greenberg, J.: Seasonal and interannual variations in whole-ecosystem BVOC emissions from a subtropical plantation in China, Atmos. Environ., 161, 176–190, https://doi.org/ 10.1016/j.atmosenv.2017.05.002, 2017.

Opacka, B., Müller, J.-F., Stavrakou, T., Bauwens, M., Sindelarova, K., Markova, J., and Guenther, A. B.: Global and regional impacts of land cover changes on isoprene emissions derived from spaceborne data and the MEGAN model, Atmos. Chem. Phys., 21, 8413–8436, https://doi.org/10.5194/acp-21-8413-2021, 2021.

Orlando, J. J., Nozière, B., Tyndall, G. S., Orzechowska, G. E., Grazyna, E., Paulson, S. E., and Rudich Y.: Product studies of the OH- and ozone-initiated oxidation of some monoterpenes, J. Geophys. Res., 105, 11561–11572, https://doi.org/10.1029/2000JD900005, 2000.

Palmer, P. I., Jacob, D. J., Fiore, A. M., Martin, R. V., Chance, K., and Kurosu, T. P.: Mapping isoprene emissions over North America using formaldehyde column observations from space, J. Geophys. Res.-Atmos., 108, 4180, https://doi.org/10.1029/2002JD002153, 2003.

Stavrakou, T., Müller, J.-F., Bauwens, M., De Smedt, I., Van Roozendael, M., and Guenther, A.: Impact of Short-Term Climate Variability on Volatile Organic Compounds Emissions Assessed Using OMI Satellite Formaldehyde Observations, Geophys. Res. Lett., 45, 8681–8689, 2018.

Wang, H., Wu, Q. Z., Guenther, A. B., Yang, X. C., Wang, L. N., Xiao, T., Li, J., Feng, J. M., Xu, Q., and Cheng, H.: A long-term estimation of biogenic volatile organic compound (BVOC) emission in China from 2001–2016: the roles of land cover change and climate variability, Atmos. Chem. Phys., 21, 4825–4848, https://doi.org/10.5194/acp-21-4825-2021, 2021.

Zhang, M., Zhao, C., Yang, Y., Du, Q., Shen, Y., Lin, S., Gu, D., Su, W., and Liu, C.: Modeling sensitivities of BVOCs to different versions of MEGAN emission schemes in WRF-Chem (v3.6) and its impacts over eastern China, Geosci. Model Dev., 14, 6155–6175,

https://doi.org/10.5194/gmd-14-6155-2021, 2021.

*3. The ozone and SOA from BVOCs due to vegetation biomass variability increase during 1981-2018. I can't understand why they will not show the increasing trend. Because almost all the model inputs are the same, but the vegetation biomass increase. The scenario settings are not so reasonable, and that weaken the significance of the results to policy management. Two real historical simulations should be better.*

**Response:** Thank you for your suggestion. The vegetation change is the main driver of interannual variations of BVOC emissions (Li et al., 2020; Wang et al., 2021). The large-scale afforestation activities in recent years lead to the rapid increase of vegetation leaf biomass and therein BVOC emissions. In our study, we aim to explore the impacts of interannual BVOC emission variations on $O_3$ and SOA formation caused by vegetation biomass variability during 1981–2018. But we are sorry for the unclear statement of the increasing trend of $O_3$ and SOA from BVOCs impacted by vegetation biomass increase in the study.

During 1981–2018, due to the changed BVOC emissions caused by vegetation leaf biomass variability, both $O_3$ and SOA formations showed significant increasing trends (p<0.05 at the 95% confidence interval) by t-test. In the revised manuscript, line 344, "At the 95% confidence level, the increasing trend for $O_3$ is significant." is added. Line 389–290, "The national SOA enhanced at an annual rate of 0.01 μg m$^{-3}$." is revised to "The national SOA enhanced at an annual rate of 0.01 μg m$^{-3}$, showing a significantly increasing trend (p<0.05).". Line 438–439, "The interannual variation of BVOC emissions caused by increasing leaf biomass results in $O_3$ and SOA concentrations increasing at average rates of 0.11 ppb yr$^{-1}$ and 0.008 μg m$^{-3}$ yr$^{-1}$, respectively." is revised to "The interannual variation in BVOC emissions caused by increasing in leaf biomass results in significant increases (p<0.05) of $O_3$ and SOA concentrations at average rates of 0.11 ppb yr$^{-1}$ and 0.008 μg m$^{-3}$ yr$^{-1}$, respectively.".

Because our study mainly explores the sensitivity of $O_3$ and SOA generation by BVOC emissions to changes in vegetation leaf biomass, all the model inputs are the same except the increased vegetation biomass. But our results can conclude that the BVOC emissions caused by vegetation leaf biomass variability is an important factor affecting the generation of $O_3$ and

SOA. This study clarifies the impact of changes in vegetation leaf biomass and is of great significance for future researches on vegetation management and precise prevention and control of air pollution in China in the context of fighting climate change. But, as you addressed, it is critical to do real historical simulations using annual meteorology and anthropogenic emissions to understand the multivariate effects on $O_3$ and SOA formation. This is explained in line 454–455, "Future work can update the anthropogenic emission inventory and use dynamic meteorological data to explore multivariate effects and provide more accurate data for evaluating the roles of biogenic emissions in air quality.".

References:

Li, L., Yang, W., Xie, S., and Wu, Y.: Estimations and uncertainty of biogenic volatile organic compound emission inventory in China for 2008–2018, Sci. Total. Environ., 733, 139301, https://doi.org/10.1016/j.scitotenv.2020.139301, 2020.

Wang, H., Wu, Q. Z., Guenther, A. B., Yang, X. C., Wang, L. N., Xiao, T., Li, J., Feng, J. M., Xu, Q., and Cheng, H.: A long-term estimation of biogenic volatile organic compound (BVOC) emission in China from 2001–2016: the roles of land cover change and climate variability, Atmos. Chem. Phys., 21, 4825–4848, https://doi.org/10.5194/acp-21-4825-2021, 2021.

*4. For the part of meteorological evaluation: using a table should be more clear.*

**Response:** Thank you for your valuable suggestion. We add a table as Table S1 in the revised manuscript to show the statistical parameters of WRF simulation for metrological variables.

Correspondingly, line 93–95, "For 2008, the average mean bias (MB), mean absolute error (MAE), and root-mean-square error (RMSE) are 0.36, 2.47, and 3.30 K over China. For 2018, these statistics are 1.24, 2.46 and 3.30 K, respectively. The correlation coefficients between simulations and observations are 0.82 and 0.86 for the year 2008 and 2018, respectively. In general, the WRF simulation is considered reasonable for driving MEGAN." is revised to "We conducted the statistical verification of meteorological variables, as shown in Table S1,

including the average mean bias (MB), mean absolute error (MAE), and root-mean-square error (RMSE). The results show that the WRF simulation is considered reasonable for driving MEGAN.".

**Table S1.** Statistics of meteorological and air quality variables.

| Variable | Year | Mean | | MB | MAE | RMSE |
|---|---|---|---|---|---|---|
| | | Observation | Simulation | | | |
| T2 (K) | 2008 | 295.84 | 295.48 | 0.36 | 2.47 | 3.30 |
| | 2018 | 295.83 | 294.60 | 1.24 | 2.46 | 3.30 |
| MDA8 $O_3$ (ppb) | 2018 | 58.83 | 36.39 | 22.44 | 36.65 | 45.42 |
| $PM_{2.5}$ ($\mu g\ m^{-3}$) | 2018 | 29.29 | 29.81 | -2.71 | 21.31 | 30.51 |

T2: temperature at 2 m; MB: mean bias; MAE: mean absolute error; RMSE: root mean square error.

*5. Fig. 2: using blue/red color table should be better.*

**Response:** Thank you for your suggestion. In the revised manuscript, Fig. 2 is revised.

[Figure]

**Fig. 2.** Spatial distribution of interannual variations in BVOC emissions caused by leaf biomass changes.